# Workforce Experiences of a Rapidly Established SARS-CoV-2 Asymptomatic Testing Service in a Higher Education Setting: A Qualitative Study

**DOI:** 10.3390/ijerph191912464

**Published:** 2022-09-30

**Authors:** Holly Blake, Sarah Somerset, Ikra Mahmood, Neelam Mahmood, Jessica Corner, Jonathan K. Ball, Chris Denning

**Affiliations:** 1School of Health Sciences, University of Nottingham, Nottingham NG7 2HA, UK; 2NIHR Nottingham Biomedical Research Centre, Nottingham NG7 2UH, UK; 3School of Medicine, University of Nottingham, Nottingham NG7 2UH, UK; 4Executive Office, University of Nottingham, Nottingham NG7 2RD, UK; 5School of Life Sciences, University of Nottingham, Nottingham NG7 2UH, UK; 6Biodiscovery Institute, University of Nottingham, Nottingham NG7 2RD, UK

**Keywords:** workforce, higher education, universities, COVID-19, SARS-CoV-2, careers, employability

## Abstract

The aim of the study was to explore workforce experiences of the rapid implementation of a SARS-CoV-2 asymptomatic testing service (ATS) in a higher education setting during the COVID-19 pandemic. The setting was a multi-campus university in the UK, which hosted a testing service for employees and students over two years. Qualitative semi-structured videoconference interviews were conducted. We contacted 58 participants and 25 were interviewed (43% response rate). Data were analysed thematically. The analysis produced four overarching themes: (1) feelings relating to their involvement in the service, (2) perceptions of teamwork, (3) perceptions of ATS leadership, (4) valuing the opportunity for career development. Agile and inclusive leadership style created psychological safety and team cohesion, which facilitated participants in the implementation of a rapid mitigation service, at pace and scale. Specific features of the ATS (shared vision, collaboration, networking, skills acquisition) instilled self-confidence, value and belonging, meaningfully impacting on professional development and career opportunities. This is the first qualitative study to explore the experiences of university employees engaged in the rapid deployment of a service as part of a pandemic outbreak and mitigation strategy within a higher education setting. Despite pressures and challenges of the task, professional growth and advancement were universal. This has implications for workforce engagement and creating workplaces across the sector that are well-prepared to respond to future pandemics and other disruptive events.

## 1. Introduction

The outbreak of severe acute respiratory syndrome coronavirus 2 (SARS-CoV-2), a strain of coronavirus that causes coronavirus disease 2019 (COVID-19), was declared a pandemic by the World Health Organization in March 2020 [1] and has posed a significant threat to the performance and viability of organisations across the globe. Sharp declines in economic activity and upheavals of labour markets (OECD, 2021) have resulted in millions of job losses [2], and drastically changed the nature of work [3,4], leading to significant stress and uncertainty for employees [5]. Such rapid organisational changes in policy and practice can provide openings for the establishment of new working environments, new services, teams, and job roles, greater cross-collaboration, more coordinated, integrative, and agile working. However, the impacts of pandemic-related transitions on career development, productivity, collaboration, and culture for the higher education workforce remains unclear.

In higher education, the COVID-19 pandemic has brought about a demand for institutions to be adaptive, and resilient [6]. Globally, this has included diverse pedagogical responses to COVID-19 [7], with many institutions transitioning to online, and blended learning models required high levels of flexibility, speed of action, and greater collaboration between institutions to maintain academic standards and high-quality learning experiences for students [8,9]. Remodelling of employment practices has been widespread, with universities extending their flexible working provision to harness opportunities presented through remote and hybrid working and meet the changing expectations of the workforce [10]. Despite the recognised challenges of home working, remote work and virtual meetings will undoubtedly persist post-pandemic [11] with only 6% of people wishing to return to full-time office working [12].

Universities across the world have implemented mitigation strategies for managing SARS-CoV-2 transmission associated with higher education, including strategies for assessment of risk in certain environments (e.g., student households, teaching and learning facilities, social environments), phased re-openings, behavioural and environmental interventions, testing, contact tracing and support for social isolation (e.g., [13,14,15,16,17,18]). One mitigation approach is routine surveillance testing for asymptomatic persons on university campuses, which, in combination with behaviour-based interventions, has shown to be feasible, acceptable [19,20,21,22] and successful in identifying infections, halting or reducing onward transmission and reducing total caseload [15,23,24,25,26].

The University of Nottingham (UoN) was one of the first institutions in the United Kingdom (UK) to establish an asymptomatic SARS-CoV-2 testing service (ATS), at pace, early in the pandemic, that demonstrably prevented COVID-19 outbreaks through early identification and isolation of positive asymptomatic cases [19,27]. In July 2021, UoN became the first institution in the UK to gain accreditation status from the oversight body, UKAS (UK Accreditation Service)—this allowed results to be reported directly to the government organisation, Public Health England which required those testing positive for SARS-CoV-2 to adhere to national laws [27]. The deployment and timeline of the ATS is contextualised in terms of tests delivered and the changing pandemic circumstances in Figure 1.

The UoN ATS has shown to be feasible and acceptable to university staff and students at various points in the pandemic, following deployments of testing across multiple campuses [19,21] and targeting university residences [20]. These studies explore the views of staff and students towards the ‘availability’ and ‘implementation’ of this service in a higher education setting in terms of preventing and managing virus outbreaks. However, the impacts on the workforce associated with establishing and operationalising new services during a time of global crisis has not been explored.

Therefore, the aim of the study was to explore workforce experiences of the rapid implementation of an ATS in a higher education setting during the COVID-19 pandemic. The research questions were (1) Did involvement in the ATS influence job roles and/or career development of personnel engaged in strategic or operational activities? (2) What features of the ATS meaningfully contributed to the relationship between ATS involvement and workforce experiences?

## 2. Materials and Methods

### 2.1. Study Design

A qualitative design [28], nested within a process evaluation of an asymptomatic COVID-19 testing service (ATS), was used. The study aimed to explore participants’ views and experiences of their involvement in the operationalising of the ATS. Data were collected using one-to-one, semi-structured, telephone or video-conferencing interviews. The semi-structured interview schedule (Appendix A) was developed by the research team, composed of a health psychologist, workforce researchers, and qualitative methodologists. The questions were pilot tested prior to study start. Although interviews involved broader questioning, to focus this paper, we have chosen to interrogate data only from question items related to specific workforce impacts of service involvement; views towards the implementation of the service will be explored elsewhere. This study complies with the Declaration of Helsinki. The service is registered with UKAS (Reference: 307727-02-01) and the process evaluation was pre-registered on ClinicalTrials.gov in September 2021 (Identifier: NCT05045989). Mixed-methods evaluation of service has been published [19,20,21]. This study was approved by the University of Nottingham Faculty of Medicine and Health Science Research Ethics Committee (REC reference: FMHS 96-0920).

### 2.2. Sampling and Recruitment

The study was conducted at a multi-campus university in East Midlands, UK. All individuals who were involved in the operationalising of the ATS were eligible to take part (the ‘ATS workforce’, *n* = 58) and were sent information about the qualitative study. The key roles and teams within the ATS are summarised in Figure 2. The total pool of ATS staff included academics (*n* = 5), project and quality management (*n* = 5), laboratory staff (*n =* 22, laboratory assistants, monitors, team leads, and technicians), information technology, software automation or project analysts (*n* = 5), sample coordinators (*n* = 20, support assistants, coordinators, set-up team) and communications officer (*n* = 1). The ATS workforce was diverse, including individuals with more, or less, experience of major project work, and those from scientific and non-scientific backgrounds. ATS personnel included senior leaders, administrators and managers, mid-career, and early career staff (e.g., pre-, and post-doctoral researchers), staff seconded from other roles in the university or the National Health Service (NHS), and staff who delayed retirement to support the pandemic response. Those who were interested in participating provided written informed consent online. Consenting participants were given the option to participate in the interview either by telephone or by videoconferencing, at a mutually convenient date and time.

### 2.3. Data Collection

The interviewers were two female health researchers trained in Good Clinical Practice (GCP) [29] and research interview skills (IM, NM) [30]. They had no prior involvement with the ATS. Interviews were conducted between May and July 2022. For national context, this follows a period of dominance of the Omicron BA.2 variant in the UK from Jan–May 2022, after which the Omicron BA.5 variant became the more prevalent, and subsequently the dominant variant by August 2022. Interview duration ranged from 24–80 min (mean: 44 min). Interviews were audio-recorded, transcribed verbatim, checked for accuracy against the original audio files, and anonymised with pseudonyms and replacement words for people and places to maintain confidentiality. Through a process of respondent verification, one of the participants listened to nine (36%) of the interview recordings (including their own recording) and checked transcripts for accuracy.

### 2.4. Analysis

Qualitative data were analysed using Braun and Clarke’s (2006) [28] six-phase guide for inductive thematic analysis. A senior qualitative researcher (SS) read and coded all transcriptions using NVivo (Release 1.0) [31]. Two other members of the research team (IM, NM) reviewed the coding for the whole dataset, including the first sweep of initial coding, and determination of the key themes. A subset of 30% of the transcripts were then analysed independently (by IM and NM), to confirm the themes [32]. The final themes were agreed through discussion between four researchers (SS, IM, NM, HB) to ensure they reflected the most pertinent issues raised in the interviews. The final analysis reported is based on the combined interpretation of the data.

## 3. Results

A total of 58 employees were invited to participate in an interview—these were all staff listed as ATS personnel at the time of the study. Of these, 25 consented to take part in the study (men: *n* = 8, 32%; women *n* = 17, 68%); 1 declined and 32 did not respond. Interview participants were aged 26 to 63 years (mean = 40.72, s.d. = 12.64). Table 1 shows a summary of participant characteristics.

The analysis produced four overarching themes (Figure 3): (1) feelings relating to their involvement in the service, (2) perceptions of teamwork, (3) perceptions of ATS leadership, (4) valuing the opportunity for career development.

### 3.1. Feelings towards Involvement in the ATS

Overall, participants were very positive about their involvement in the service:


*I wouldn’t change anything about it. It’s all highly positive from my perspective.*
(Participant 19)

For some, the opportunity to be involved had instilled a sense of pride at being able to contribute towards the prevention and management of outbreaks of COVID-19 on the University’s campuses.


*I think it was… a hugely valuable thing that the university put on for staff and students…and I’m very proud to have been a part of it.*
(Participant 24)


*I would do it again, at the drop of a hat. I just think it was, it was a fantastic service that the university set up…and very proud to have been a small part of that.*
(Participant 20)


*very proud, to be, part of it. Something that’s worked so well. And that’s been so useful.*
(Participant 25)

This was rewarding to participants who felt they were able to help people at a time when nationally and globally, there was a high level of uncertainty relating to the pandemic.


*It has been a great honour…at the end of the day you are helping keep people safe.*
(Participant 15)

At an individual level, for participants who had been on furlough from their main job role during the pandemic (a period of temporary leave), the opportunity to be involved in the ATS gave them a sense of purpose, keeping them engaged and motivated at work.


*they wanted to contribute something in the pandemic and not be furloughed and sitting at home.*
(Participant 18)


*I was pleased to be of some use, actually… I think I would’ve hated it if I’d …not done anything that was of direct use in the pandemic.*
(Participant 23)

Participants reported feeling valued by their organisation, and this was fundamental to their job satisfaction and work engagement.


*there are some good lessons to be learnt …we didn’t have micromanaging, people felt valued by the management, people felt they were on board…I think the value meant that they took the responsibility and got on with the job.*
(Participant 4)

Most of the participants felt stressed or under pressure at some point during their time on the ATS. For some this was simply the uncertain nature of the working environment in the context of a pandemic, and for others it was the challenge of balancing their role on ATS alongside other commitments.


*Yes, to be honest I couldn’t tell you how I did it. It just got done, it was very much, that is the only thing, I think because everything was so fast you didn’t have time to stress over if this is correct or you know you have don’t have time to sit and worry, you just have to get it done and that almost forces you to just get on.*
(Participant 1)

One participant reported a sense of frustration with the delays to accreditation of the service due to uncertainties about the processes involved: 


*accreditation took too long and was too complicated… that was just massively delayed…I think there was an easier route but again if you don’t know policies and you don’t know what you are doing, it is going to take time.*
(Participant 1)

There was a sense that those who were new to major projects, or the testing environment struggled at times due to their lack of prior experience. Those with previous experience in research or projects involving delegation of responsibilities appeared more readily able to adapt and cope in the fast-paced ATS environment.


*I think the good thing is that a lot of people had experience with kind of projects similar to that…so that really helped to organise things a lot better … having people that actually knew how to handle projects like that, I think made a massive difference to how well it worked…*
(Participant 21)

Despite some reports of stress, pressure, and challenges due to the rapidity of effort required, the necessary urgency of service set-up and delivery was described by participants as one of the key reasons the implementation of the service was a success: *I think the speed and attitude towards, of the staff trying to get the service up and running correctly, made it the success that it is.* (Participant 14)

### 3.2. Perceptions of Teamwork

All participants in the study reported working with new people and new team(s) during their involvement with ATS. Staff members working on the ATS brought various skills acquired within their previous roles, such as logistics, research project management, laboratory testing, etc.


*There was a lot to it, it was very complicated, having to deal with lots of different types of people in something that they may not have had experience in beforehand…*
(Participant 21)

Working with new team members and individuals from different departments brought with it some inherent challenges, especially given the *“**storming, forming, norming” kind of phase that you go through with team development* (Participant 16).

There was suggestion from some participants that those staffing the ATS were chosen in part for their ability to work well within a team; this recruitment strategy may reflect why so many participants reported a positive experience of teamwork on the project.


*we were looking for people that could work well within a team and integrate well, and I think we managed to, to find and recruit really beautiful team players. And…pretty much everybody in the team was very good [laughs] at working with people, they integrated very well, they were very cooperative, collaborative, hard, working, communicative.*
(Participant 9)

This positivity towards the team may also be attributed to a shared goal: 


*‘the thing that helped us all was that we all had a very focussed objective that was business critical and so everybody bought into that, and everybody wanted to deliver it so we came together as a team very quickly.’*
(Participant 16)

One participant alluded to ‘all being in it together’, suggesting that equality helped facilitate teamwork: 


*‘we all just mucked in and got on with what needed to be done and I think everybody acknowledged that different people had different levels of expertise…it was just solely let’s roll our sleeves up, let’s get on with it’.*
(Participant 4)

It was this likeminded attitude and having a common goal that contributed to effective teamwork and making the service a success in the eyes of its staff.


*it wouldn’t have gone as well, if it didn’t have that same team and everybody open to change, everybody open to just getting the service up and running as good as we can, as professional as we can to keep the students safe and to keep the staff safe.*
(Participant 19)

As with leadership, communication was considered a key facilitator of effective teamwork within the ATS. Communication was often perceived to be more effective within the ATS teams, compared to participants’ prior or substantive roles.


*Very different to what I am, what I have been used to, but in a really positive way. It very much was a team effort from top to bottom. And it’s the communication and always working, transparent and open, not just with the customer but with ourselves that I think made it a really strong and unique team as well.*
(Participant 5)

Effective team communication was critical to efficient and smooth running of the service. This was facilitated by technology—the use of video-conferencing platforms enabled participants to rapidly identify problems and challenges and facilitate expedient solutions during the rapidly changing pandemic context.


*because of, having communication via Teams…it’s pretty instant, and everybody’s, you know, sort of hot on it, you know what I mean? There hasn’t been any delays in any communicating anything.*
(Participant 22)

Participants’ views on teamwork within ATS were influenced by the level of support they received either from the leadership or from colleagues within their teams.


*people were learning a little bit on the job and learning how to do managerial roles and they needed some, some support and guidance, but, but in the main, I think the, the level of cohesion and support was, was very, very strong. And I think that’s been one of the things that’s made it a big success.*
(Participant 8)

### 3.3. Perceptions of ATS Leadership

The interviews demonstrated that most participants had a positive view towards the ATS leadership with a key feature being ‘flattening of the hierarchy’ and perceived equality.


*…there has always been strong leadership throughout my time with the service… there was no kind of hierarchy, it wasn’t ‘this is your senior manager and this person you can’t speak to them’, it was very much ‘we’, from every level, everybody was working towards the same goal, and we were all working together …*
(Participant 5)


*We are not hierarchical, everyone is allowed to participate in the conversations, to put their views forward for any discussions and then we all work towards a solution to any issues.*
(Participant 16)

Several participants indicated they appreciated a hands-off approach to leadership, and a lack of micro-management which engendered trust, and allowed individuals to step-up and take responsibility.


*it’s been very… hands off in some ways. We’ve been just, to get on with things—which has been nice.*
(Participant 14)

However, this approach did not suit all. For some participants, the lack of hierarchy was challenging. There were smaller teams within the ATS in which there was no designated individual to take responsibility, which hampered decision-making.


*I felt like we have all needed a bit of direction and someone to take authority and make final decisions when maybe we are not able to come to a decision ourselves and when we are a bit inconsistent in our approaches and things like that. I have not felt that we have had that direction from above…*
(Participant 17)

Positive comments relating to the ATS leadership were often related to the practice of being mindful and compassionate towards one another, which in turn facilitated a constructive working environment.


*It was just really, really, supportive, and really open to everybody’s opinion; you felt really valued…*
(Participant 13)


*…they promoted a good work environment and a good team environment.*
(Participant 9)

Many participants noted the importance of good communication operating in conjunction with strong leadership. The approachability of the senior leadership team coupled with collaborative communication styles and regular exchange of information helped to make a positive impact on staff.


*… without this support, without this constant communication with the leadership, I don’t think … we would have been able to achieve half of what we did. So, what I really appreciated was the constant communication and the openness to be able to talk about anything and that everything was given their attention and addressed.*
(Participant 9)


*I think that was the biggest thing that helped, that, someone being approachable and relatable. I think in any leadership role, but especially this, as it was so fast paced, everyone was under a lot of stress.*
(Participant 21)

Overall, most participants valued the approachability of the leadership team, the perception of a horizontal management structure and the direct involvement of team members in decision-making and problem-solving. The leadership team were perceived to have created a psychologically safe environment in which people did not fear making mistakes and felt able to speak up.

### 3.4. Valuing the Opportunity for Career Development

All the participants appreciated the impact that their involvement in the ATS had on their personal development and/or career progression.


*Other than, yeah, developing the skills and getting the promotion, it sort of gave me a jump start for a career that I didn’t know was a thing. So, I guess, it started my career in clinical research a bit more, it gave me a massive jump start anyway.*
(Participant 1)

Most of the participants acknowledged either gaining new transferable skills or improving existing ones during their role on the project, which had opened up new career opportunities.


*being able to collaborate and communicate with different teams also kind of relationship management, negotiation skills as well has definitely been strongly developed from my time in the service… If I hadn’t done that, I don’t think I would have developed those skills for another number of years.*
(Participant 5)


*So, I think it has helped a lot of people with their careers, it’s added value to and completely new things, they hadn’t even considered, so yes, it’s good, technical and IT information on their CV.*
(Participant 6)

All participants felt their experience on the ATS project helped them in the immediate to medium term within their career trajectory. Participants valued the opportunity to undertake further training regardless of their position within the service that would enhance their job prospects in the future.


*I was very pleasantly surprised that at my level, I was offered PRINCE2 [project management] training. I did it straight away, and within two weeks…I’m now a licensed PRINCE2 practitioner—which has helped a lot in [job] applications…it has helped me kind of understand that project management is something that I want to do.*
(Participant 21)


*the thing which was a bit newer to me and I definitely got to, like, develop through this job was kind of managing large databases…we have all seen an excel file but mine were a lot smaller, you know like 50 lines or something, like small, small, whereas with this we had data from like thousands and thousands of people…*
(Participant 2)

Most participants also felt that their involvement had a long-term impact on their career, through the opportunities to collaborate with people from other disciplines (and developing the confidence and skills to do this effectively), and the broader networks they had created.


*… how to talk to people that are higher up than you, and in a confident way. I think that was a big one. How to talk with lots of different people from different skills and how to collaborate… it was a challenge, but you have got to learn it someday haven’t you if you want to, if you want to climb the ladder.*
(Participant 1)


*…probably give me some, you know, better or high opportunities … because I will have had two years of valuable experience and in a role that otherwise I would not have had.*
(Participant 3)

Most of the participants reported that they received a promotion during their role within the ATS or immediately afterwards either on return to their prior roles, or they had been appointed at a higher level when applying for new posts: *I got promoted soon after I came back into a, into a managerial role within my team* (Participant 19). Involvement in the service was invaluable to these participants for their career development giving them a head start compared to their peers.


*I have moved up two levels so previously I was level 5, and I am about to take on a level 7 role, which is quite quick considering I was level 5 less than 2 years ago and that’s all, 100% down to the experience I had in ATS that unlocked my confidence, my ability to talk about what I do more openly, my acceptance that I have value to the organisation, all that kind of stuff was hidden before.*
(Participant 16)

## 4. Discussion

To our knowledge, this is the first study to explore the workforce impacts of involvement in the implementation of a SARS-CoV-2 testing service established to prevent and manage virus outbreaks in a higher education setting. The key themes are distinct, yet interrelated, focusing on participants’ broader feelings towards their involvement in the service, perceptions of ATS teamwork and leadership, and opportunities for career development. Overall, participants in this study communicated positive views towards their involvement in the ATS, demonstrating prosocial values through a willingness to be engaged in, and contribute to, the University’s response to the COVID-19 pandemic through service implementation. Such prosocial attitudes were also identified in university students as a reason for uptake of the SARS-CoV-2 testing being offered by the ATS [33].

This study demonstrates the significant career development opportunities afforded by the establishment and implementation of the ATS during the pandemic, across levels of employment and job type. Structured career development opportunities may have been reduced or side-lined during the crisis, in favour of instigating changes, streamlining operations, cutting costs and organisational ‘survival’. However, in higher education, COVID-19 expedited changes within weeks that may otherwise have taken decades [34]. This paved the way for involvement of individuals in the establishment of innovative new services, and pandemic response strategies requiring new (and rapid) approaches which generated potential to foster the development of personal skills in the workplace. Our participants described their involvement in the ATS as a catalyst for significant personal and professional development, resulting in recognition, reward, and promotion.

Skills acquisition has broad value for employability since the type of skills that are most valued by employers has recently changed. For example, an analysis of the skill profile of the most-in-demand jobs during the peak of the COVID-19 crisis demonstrates a shift in skills demand during the pandemic, with employers increasingly looking for technical and transversal skills [35]. With relation to technical skills, many of our interviews developed skills relating to overall project management and budgeting, data handling and analysis, information technology (IT) and computing (e.g., word processing, spreadsheets, databases, presentations, design skills), and/or laboratory skills (e.g., record keeping, safety, risk management, pipetting, sterilisation, equipment calibration and maintenance, SARS-CoV-2 testing, etc.). The nature of technical skills varied by job role, for example, many early career researchers and scientific staff developed or enhanced laboratory skills; administrators and managers commonly developed or enhanced project management, data and IT skills. Due to the ongoing pandemic and its sequalae, there is likely to be a continued growth in the demand for workers in healthcare and science, technology, engineering, and mathematics (STEM) occupations, and a need for individuals with the skills to create, deploy, and maintain new technologies [11]. UNESCO defines transversal skills as ‘skills that are typically considered as not specifically related to a particular job, task, academic discipline, or area of knowledge and that can be used in a wide variety of situations and work settings’ [36]. These relate to the capacity of individuals to problem solve and take initiative, communicate with others (e.g., provide and receive precise instructions under pressure), and work effectively in teams; skills that are needed to predict successful careers, retain talent in organisations and are closely connected to employability [37,38]. In 2017, Deloitte Economics forecasted that two-thirds of all jobs by 2030 will comprise transversal skills-intensive occupations (‘soft-skills’) [39], requiring leadership, teamwork, flexibility and adaptability, critical thinking, self-management and professional ethics, digital skills, communication and emotional intelligence, creativity, and innovation. These skills were developed within the ATS teams across all job categories, and will be increasingly valued by employers in the changing context of work during and after the COVID-19 pandemic [35].

The COVID-19 pandemic has been (and remains) a highly disruptive event, and organisations globally have altered policy and practices that have led to pandemic-enforced workforce shifts in job roles, ways of working and communicating. The ATS is one example of this. For some participants, personal reflection and re-framing of career goals led to anticipated or actual career change. At an individual level, the need to negotiate unpredictable events in a rapidly changing career environment may be conceived of as a ‘career shock’, a concept that arose in the 1990s [40] but has more recently been applied in the context of COVID-19. A career shock can be defined as “a disruptive and extraordinary event that is, at least to some degree, caused by factors outside the focal individual’s control and that triggers a deliberate thought process concerning one’s career” [41]. Individual career shocks can have widespread ramifications for employing organisations as people reassess their current positions and anticipated future situations. In higher education, many employees have been worried about their career development during the pandemic and considered leaving the workforce or reducing their hours [42]. Our findings demonstrate the value of the ATS in creating career development opportunities that may not have been available to some individuals (certainly as quickly) pre-pandemic. Some of our participants had been furloughed (placed on temporary leave) prior to their involvement in the ATS. Such changes in job status during the COVID-19 pandemic are purported to impact significantly on employees’ mental wellbeing [43]. Since the ATS provided an opportunity for reallocation of skills within the organisation, furloughed participants in our study were able to remain active contributors to the higher education workforce.

As discussed by Akkermans and colleagues [44], for many, the impacts of the COVID-19 pandemic on individuals’ careers are commonly negatively valanced (capturing ‘bad’ emotions) associated with, for example, the challenges of pandemic-related uncertainty, anxiety, loss of income, or job insecurity. However, our study demonstrates that career impacts can also be positively valanced (capturing ‘good’ emotions) since the disruptive event (pandemic) provided opportunities for change in work environments, upskilling and proactive career behaviours.

While organisational change may have negative impacts on individuals (e.g., work-related stress, burnout: [45]), particularly during times of change [46], these impacts can be moderated by job-related factors (e.g., greater job control) and supportive leadership styles [47,48]. This was evident in our sample. First, participants had self-nominated to a role within the ATS, and perceived that within their role, they had some level of influence and control (e.g., reports of perceived equality, ‘flattened hierarchies’, autonomy, and team decision-making). Second, the vast majority spoke positively of the supportive and inclusive leadership styles within the ATS. Most notably, the approachability of the leadership and engaging staff in problem-solving and decision-making instilled a sense of team cohesion, facilitated timely, honest, and transparent communication, and engendered psychological safety (“a shared belief held by members of a team that the team is safe for interpersonal risk taking”). The value of inclusive leadership and the mediating role of psychological safety in curbing unwanted adverse outcomes (e.g., psychological distress) has been demonstrated elsewhere (i.e., public sector workers: [49]). This is relevant since work-related stress and worsening mental health was evident in the higher education sector prior to the pandemic, with growing concerns for staff wellbeing during and beyond it [50]. Although most participants in our sample reported experience of work-related pressures associated with their involvement in the ATS, pressure and stress appeared to be mitigated by the rewarding and positive aspects of their role within the ATS (e.g., cohesive teams, significant job satisfaction, skills acquisition, high work engagement and productivity, and career development opportunities). The leadership style described by our participants involved a ‘flattening of the hierarchy’ with regular, reciprocal, and co-operative communication. Such adaptive leadership styles have been recognised as important for ensuring the continued functioning of academic institutions during the COVID-19 pandemic [51].

Our study identified that a major impact of the ATS on the workforce (and the subsequent success of the ATS) was effective teamwork. The COVID-19 pandemic is an era of fast-paced teaming; a flexible approach to teamwork in which essential collaborators are identified, quickly getting up to speed on what they know so they can work together on a fast-moving issue, with a shared goal. This ‘teaming on the fly’ approach involves working on a problem and concurrently seeking better approaches, thus executing, and learning at the same time [52]. This was evident in the ATS in which service continuity or adaptation was required under taxing and uncertain conditions. Participants in our study described the impact of teaming approaches in terms of the value of shared vision and goals, mindfulness towards others, and inclusivity. Our findings align with Edmondson’s three pillars of teaming culture [53] where leaders build a culture that enables curiosity (finding out what others can bring to the table), passion (fuelling enthusiasm and effort) and empathy (seeing another’s perspective).

Our participants included individuals with more, or less work experience (at the host institution or elsewhere) and included early career researchers. At times of organisational change, choices, decisions, and strategies will vary whether individuals are in the early, mid, or later stages of their careers [41]. In higher education, those in the earlier stages of their careers have expressed fear, uncertainty, and anxiety about their career prospects in the post-pandemic future [54,55]. Concerns for the future were not evident in our sample. Indeed, the ATS was seen to provide a springboard for early career researchers who gained experience of major project work and leadership skills that they may not ordinarily have acquired at this stage of their career.

Separately, we have explored the views of ATS staff towards mass testing in a higher education setting, and the barriers and enablers of service implementation to generate lessons learned for future service delivery in a time of crisis [19,20,21]. The current study focused specifically on the workforce impacts of the rapid establishment and sustained delivery of the service, reflecting on a two-year period to the point of service de-commissioning. It would be interesting to explore whether the career development benefits afforded to staff are sustained in a post-pandemic era, and whether, and how, the model of teamwork and leadership adopted in the ATS is applied by this workforce in the transition back to their prior roles, into new roles, or taken forward in new workplace initiatives, within or beyond times of crisis. Although there is a rapidly emerging evidence-base describing the impacts of the COVID-19 pandemic on the higher education sector, future studies might seek to move beyond immediate challenges, to insights gained from crisis response operations, and value added to the higher education workforce.

### Study Strengths and Limitations

This is the first qualitative study to explore the work-related experiences of university employees engaged in the rapid deployment of a service as part of a sustained dynamic pandemic outbreak and mitigation strategy within a higher education setting. Participants were recruited from across the ATS workforce, including university employees and early career researchers, providing a rich exploration of stakeholders’ experiences. Participants in this convenience sample may not represent employees delivering other types of service, or operationalising services outside of this organisation, or the pandemic context.

## 5. Conclusions

In conclusion, the ATS fostered a culture that promoted agile learning and rewarded innovation in work processes. Agile leadership and teaming were needed to ensure vital decisions were made to protect the health of the academic community and the public. Inclusive, compassionate leadership styles created a culture of psychological safety and engendered team cohesion which facilitated the implementation of a rapid mitigation service, at pace and scale. Specific features of the ATS (shared vision and decision-making, collaboration, networking, skills acquisition) empowered the ATS workforce, instilled self-confidence and a sense of value and belonging. This meaningfully impacted on professional development and career opportunities across all job categories. Despite pressures and challenges of the task, and support needs of staff with less decision-making experience, professional growth and advancement were universally reported. This has implications for staff wellbeing, work engagement, and the creation of workplaces across the sector that are well-prepared to respond to future pandemics and other disruptive events.

## Figures and Tables

**Figure 1 ijerph-19-12464-f001:**
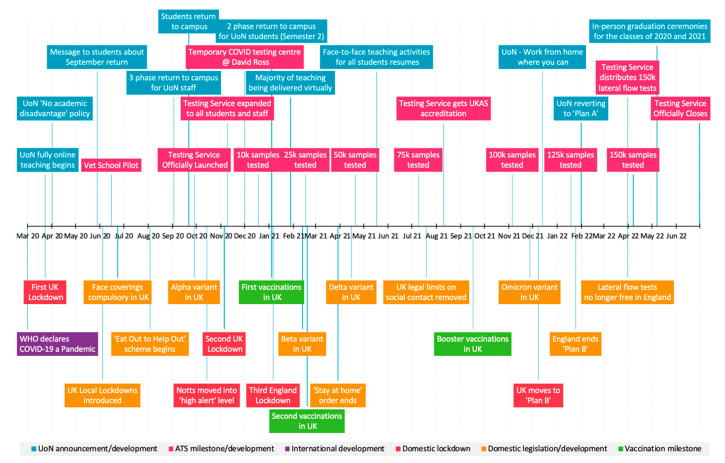
Deployment of Asymptomatic Testing Service and the national COVID-19 context. UK: United Kingdom; WHO: World Health Organization; UoN: University of Nottingham; Notts: Nottingham; Vet School Pilot [19]; David Ross: University sports facility and temporary testing site venue; Plans A and B (see www.gov.uk accessed on 28 September 2022).

**Figure 2 ijerph-19-12464-f002:**
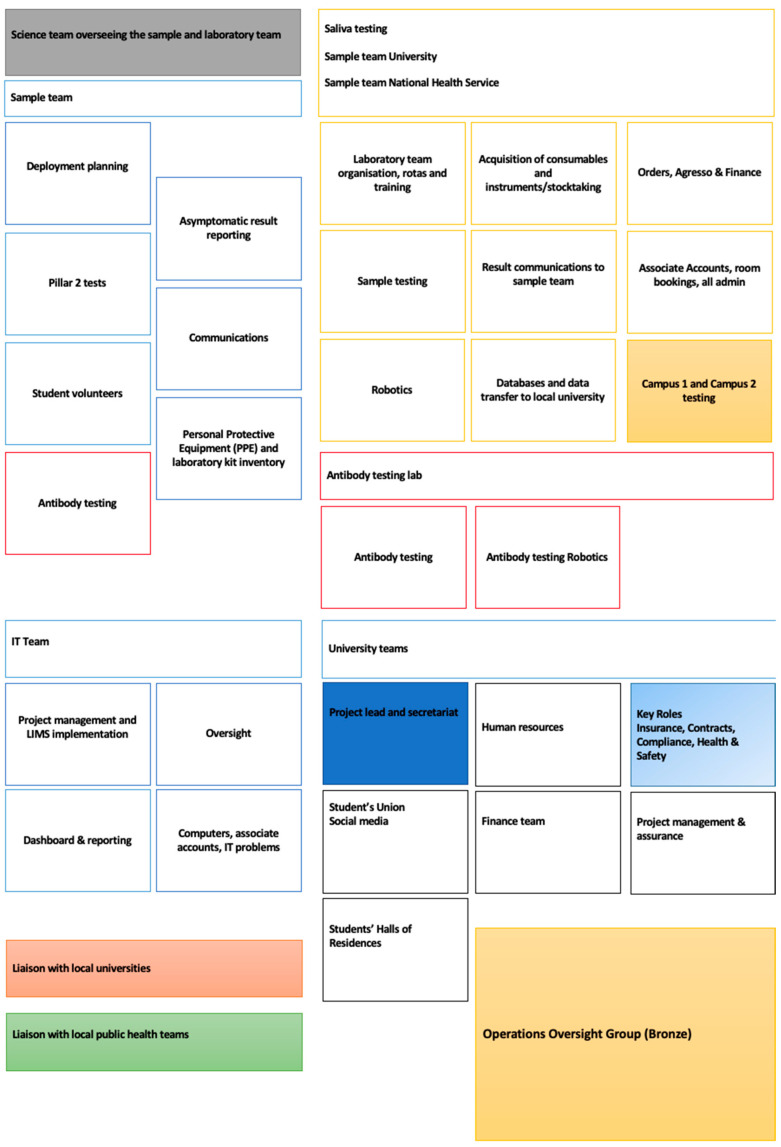
Roles and teams in the ATS. Shading: Grey: Science team (samples and laboratory work); Orange: Oversight and deployment; Rose/Green: Community liaison; Blue: Leadership/management (dark) and key governance ‘stop-go’ roles (pale). Box line colour: used to group teams into areas of work.

**Figure 3 ijerph-19-12464-f003:**
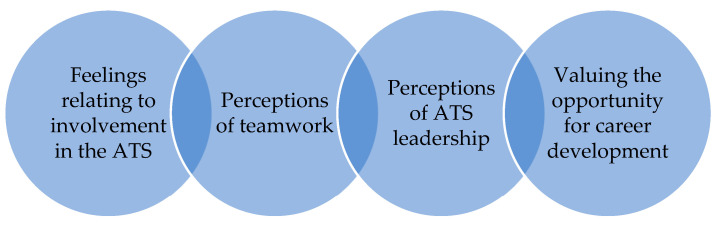
Overarching themes.

**Table 1 ijerph-19-12464-t001:** Participant characteristics.

Respondent	Gender ^a^	Age	Job Role ^b^
1	F	20–30	Research
2	F	20–30	Research
3	M	31–40	Laboratory
4	F	61–70	APM ^c^
5	F	31–40	APM
6	F	61–70	Laboratory
7	M	31–40	Laboratory
8	M	41–50	Research
9	F	31–40	Laboratory
10	M	21–30	Laboratory
11	F	31–40	Laboratory
12	M	41–50	Research
13	F	51–60	APM
14	F	20–30	APM
15	F	31–40	Laboratory
16	F	51–60	APM
17	M	31–40	APM
18	F	51–60	Laboratory
19	F	20–30	Research
20	F	41–50	Research
21	F	20–30	APM
22	F	61–70	APM
23	M	61–70	Research
24	M	31–40	Other ^d^
25	M	41–50	APM

^a^ F = female, M = male; ^b^ prior to joining the ATS; ^c^ Administrative, professional, and managerial; ^d^ Other: IT or technical.

## Data Availability

The data that support the findings of this study are available on reasonable request from the corresponding author. The data are not publicly available due to their containing information that could compromise the privacy of research participants.

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
