# Peer review of "Workforce Experiences of a Rapidly Established SARS-CoV-2 Asymptomatic Testing Service in a Higher Education Setting: A Qualitative Study"

_ijerph, 2022, doi:10.3390/ijerph191912464_

Round 1

Reviewer 1 Report

It was a pleasure for me to read this article. The topic is interesting, and the narrative flow is easy to follow. The article applies an appropriate methodology and presents the results clearly. The purpose of the study is described in a logical, comprehensible, and explicit manner.

However, the author (s) should also consider the following suggestions:

- The authors should mention the source for all figures and tables in the paper.

 2.3. Data collection

- This section should be improved with information about the research period.

3.Results

- Line 155: There is already a figure numbered as Figure 1 in the paper. This figure should be renumbered.

4. Discussion

- Line 383: Because it is the first appearance in the text, the full name for the STEM must be used and then the abbreviation can be used.

- The future directions of research were not presented. I suggest to the author(s) also address these elements if they have been identified.

5. Conclusions

- The author (s) should better highlight the implications of the findings on all stakeholders (medical staff, researchers, others). Also, perhaps the contributions (implications of this study) should be grouped into theoretical and practical contributions.

Reviewer 2 Report

This is a well-written report of a study in a higher education setting in the UK to explore the workforce experiences of the rapid implementation of a SARS-CoV-2 asymptomatic testing service. The semi-structured interview schedule is said to be found in Supplementary file 1; however, there is no Supplementary file 1. There is, on the other hand, a Supplementary file 2 which seems to be the semi-structured interview schedule. Based on the order of the questions as the appear in that file, the four overarching themes that were analyzed were 1) feelings relating to their involvement in the service, 2) perceptions of teamwork, 3) perceptions of ATS leadership, 4) valuing the opportunity for career development. The intent of the figures is helpful; although they are missing legends and Figure 3 is incorrectly labeled as Figure 1. As well, this incorrectly labeled figure also orders the four overarching themes in a way that differs from the order of questions asked to participants. As such, these themes should be reordered in the figure as suggested below in the line by line edits. Generally well-referenced, there are still a few references missing. On the whole, this appears to be an appropriately conducted and reported study.

Line by line suggested edits

19-21 Change “1) perceptions of ATS leadership, 2) perceptions of teamwork, 3) valuing the opportunity for career development, 4) feelings relating to their involvement in the service” to “1) feelings relating to their involvement in the service, 2) perceptions of teamwork, 3) perceptions of ATS leadership, 4) valuing the opportunity for career development”.

35 Change “coronavirus (COVID-19),” to “severe acute respiratory syndrome coronavirus 2 (SARS‑CoV‑2), a strain of coronavirus that causes coronavirus disease 2019 (COVID-19), was ”.

36 Need a reference for the date the WHO declared COVID-19 a pandemic.

45 Change “coronavirus (Covid-19)” to “COVID-19”.

46 Delete “even more”.

Figure 1

Please remove the title inside the figure and use that title as the title of the figure. In other words, change “Asymptomatic Testing Service and Covid-19 Timeline.” to “COVID-19 Period—Key Events”. Add a legend inside the figure to indicate what is represented by each of the colours—blue, rose, red, gold, green and purple. 

91 Change “covid” to “COVID”. Please provide a reference for the type of qualitative design used for the study.

95 Supplementary file 1 has not been included in the supplementary file received.

Figure 2 

Change “Result comms to sample team” to “Result comes to sample team”. Add a legend inside the figure indicating the meaning of the different colours, why some boxes are shaded and others only outlined, and why one of the blue shades is much darker than the other.

126 Please provide a reference for Good Clinical Practice and for the particular research interview skills used.

130 Were all participants given the option of reviewing their interview transcript or was this a request of the participant?

135 Please provide a reference for the NVivo 12 software used.

155 Change “Figure 1” to Figure 3”. If the information in Supplementary File 2 represents the order in which questions were asked of participants. In contrast, this figure doesn’t represent the order the questions were asked. To correspond with the order of questions asked, the circles, from left to right, should be “Feelings relating to involvement in the ATS”, “Perceptions of teamwork”, “Perceptions of ATS leadership”, and “Valuing the opportunity for career development”. This means the body of the subsections in the Results should be 3.1. Feelings relating to involvement in the ATS, 3.2. Perceptions of teamwork, 3.3. Perceptions of ATS leadership, and 3.4. Valuing the opportunity for career development.

156 Change “Perceptions of ATS leadership” to “Feelings relating to involvement in the ATS”

157-200 Move this information to subsection 3.3. Move the information in lines 300 to 354 from subsection 3.4. to this subsection.

250 Change “Valuing the opportunity for career development” to “Perceptions of ATS leadership”.

251-298 Move this information to subsection 3.4 and move the information in lines 157-200 from subsection 3.1. to this subsection.

299 Change “Feelings towards involvement in the ATS” to “Valuing the opportunity for career development”.

300-354 Move this information to subsection 3.1. and move the information in lines 251-298 from subsection 3.3. to this subsection.

359-360 Change “perceptions of ATS leadership and teamwork, opportunities for career development, and broader feelings towards their involvement in the service” to “feelings towards their involvement in the service, perceptions of teamwork and ATS leadership, and opportunities for career development”.

363 Change “Covid” to “COVID”.

366-376 Move to current location of information in lines 397-424 and move that information on feelings towards involvement in the ATS here.

397-424 Move to current location of information in lines 366-376 and move that information on valuing the opportunity of career development here.

414 Change “Covid” to “COVID”.

449 Change “Covid” to “COVID”.

451 Change “Covid” to “COVID”.

References 

For each of the references that are available online, please follow the MDPI style of listing— Available online: URL (accessed on Day Month Year).

As per the instructions to authors provided by MDPI, please italicize journal numbers and leave out the issue number.

48. Is this reference an article or a book? This is not clear from how it is referenced. 

Supplementary file 2

This file is not referenced in the document. There are methodological problems with the way the questions are organized in Supplementary file 2. When posing questions, the questions should be grouped by topic into separate sections and each point should ask only one question at a time. Here is an example of how Supplementary file 2 could be organized to correct these problems.

Supplementary File 2
ATS Staff Interview Topic Guides

You and your role 

·       What was your contribution to the service? (e.g., strategy, operations, advisory, academic, administrative, technical, testing delivery, student, or staff support?). 

·       How did you acquire this role?

·       How do you feel about your involvement in the service overall?  

Your team

·       Thinking about the team(s) you worked in – had you worked together before? 

·       What is your view towards the way people within the team(s) operated together? 

Team leadership

·       Thinking about team or service leadership - were you adequately supported in your role? 

·       What is your view of the service leadership (overall, and within smaller team(s) or area(s))? 

·       Were there particular leadership approaches that helped or hindered you in your role? 

·       Did you act in a leadership role yourself, and if so, how was it? 

Career development

·       Did your role or contribution to the service generate any immediate personal or professional development opportunities? If so, what were they? 

·       Did you develop any new skills? 

·       Has it had any implications for you in the medium-to-longer term?
